

# Comprehensive analysis of groundwater hydrochemistry and nitrate health risks in the Baiquan basin, Northern China

Bo Li[1,2], Di Wu[1,2], Dalu Yu[1,2], Changsuo Li[1,2], Jinxiao Wang[1,2], Liting Xing[3], Shuai Gao[1,2], Zhe Zhang[4], Mingguo Wang[4] and Yuwei Wang[1,2]

[1] 801 Institute of Hydrogeology and Engineering Geology, Jinan, China
[2] Shandong Engineering Research Center for Environmental Protection and Remediation on Groundwater, Jinan, China
[3] University of Jinan, Jinan, China
[4] Shandong Hydrogeology Engineering Geologyand Environment Geology Corporation, Jinan, China

Corresponding author
Dalu Yu, ydl801801@163.com

## ABSTRACT

Groundwater is a crucial water source and strategic resource, essential for sustaining both urban and rural livelihoods, supporting economic and social development, and maintaining ecological balance. This study investigates the hydrochemical properties and controlling factors of groundwater in the Baiquan basin (BQB) by analyzing water quality data collected during both dry and wet periods. Additionally, the suitability of groundwater for drinking and agricultural irrigation was evaluated. The findings reveal that groundwater in BQB is generally weakly alkaline and primarily consists of hard-fresh water. Although there are seasonal variations in the main ion concentrations, $HCO_3^-$ and $Ca^{2+}$ are the predominant anions and cations, respectively. Consequently, the hydrochemical type is mainly $HCO_3$-Ca·Mg type, with a secondary classification of $SO_4$·Cl-Ca · Mg. The hydrochemical composition is primarily influenced by the dissolution of carbonate and silicate minerals, as well as cation exchange processes. Additionally, it is affected by anthropogenic inputs, particularly from the use of agricultural fertilizers. The water quality assessment results indicated that all water samples are classified as either good or moderate, with a significant majority falling into the good category. Additionally, the northern section of the BQB exhibited lower entropy weight water quality index (EWQI) values during the dry season in comparison to the wet season. For irrigated agriculture, groundwater in the BQB serves as a high-quality water source for irrigation throughout both the dry and rainy seasons. Furthermore, non-carcinogenic risks are notably concentrated in the north-western and south-eastern regions of the study area. Health risks associated with nitrates in groundwater are elevated during the rainy season. Notably, non-carcinogenic risks for infants were significantly high across both seasons and substantially exceeded those for children and adults. These results provide valuable scientific insights for the management and development of groundwater resources in the BQB.

## INTRODUCTION

Groundwater is a valuable resource with advantages such as superior water quality and a stable supply compared to other sources. This makes it particularly crucial for meeting production, domestic, and agricultural irrigation needs, especially in arid and semi-arid regions where it often serves as the primary water source (*Gao et al., 2023a*; *Gueroui et al., 2024*; *Kushwah & Singh, 2024*; *Liu et al., 2021*). Moreover, the extensive availability and accessibility of groundwater play a vital role in maintaining ecological balance and supporting global human development (*Liu et al., 2018*; *Liu et al., 2024*). However, increasing population and economic growth have heightened the demand for water resources. Urbanization, along with rising industrial and agricultural water consumption, has exacerbated the imbalance between groundwater supply and demand (*Adjëï Kouacou et al., 2024*; *Chen, Zhang & Cai, 2023*; *Laonamsai et al., 2023*; *Liu et al., 2024*). Furthermore, improper extraction and use of groundwater have led to significant water quality issues, which have become a major concern in the ongoing struggle to balance groundwater supply and demand (*Abdulsalam et al., 2022*; *Zhao et al., 2024*).

In underground aquifer systems, variations in climatic conditions, topography, surface water, soil and aquifer material composition, rock permeability, and groundwater hydrodynamics from recharge to discharge areas result in differences in groundwater chemical composition (*Alvarez et al., 2020*; *Gueroui et al., 2024*; *Liu et al., 2018*; *Masoud & Abu El-Magd, 2022*; *Wang et al., 2024a*; *Wang et al., 2024b*; *Xia et al., 2022*). Analyzing the hydrochemical properties of groundwater and understanding their formation mechanisms are crucial for identifying the main factors and processes that control groundwater quality. This knowledge provides a scientific basis for the development and utilization of groundwater resources (*Kammoun, Abidi & Zairi, 2022*; *Liu et al., 2024*; *Mohammed et al., 2022*). Consequently, studying groundwater hydrochemistry is essential for the sustainable management of groundwater resources. Currently, hydrogeochemical methods such as graphical representation of hydrochemical types, ion ratios, and mineral saturation indices are the primary techniques used to study groundwater formation processes and are widely applied (*Dong et al., 2024*; *Liu et al., 2021*; *Masoud & Abu El-Magd, 2022*; *Zakaria et al., 2021*).

The evaluation of groundwater quality relies on established groundwater quality standards and is grounded in hydrochemical test data. This process involves employing appropriate evaluation methods to classify and assess groundwater quality (*Ghosh & Bera, 2023*; *Liu et al., 2023*). Objective evaluation results are essential for accurately understanding the current state of groundwater quality and for quickly identifying key indicators that influence groundwater quality levels. This understanding is crucial for the rational utilization and development of groundwater resources (*Abhijit et al., 2024*; *Ghosh & Bera, 2023*; *Liu et al., 2021*). Currently, various methods exist for assessing groundwater quality, including single-factor and comprehensive evaluation methods (*Gao et al., 2023b*; *Kumi et al., 2023*; *Liu et al., 2023*). *An et al. (2023)* assessed the water quality in the study area using both single-factor evaluation method and the Nemerow Pollution Index method and noted that the single-factor evaluation method is straightforward and can clearly highlight

water quality indicators that exceed acceptable limits. While it effectively identifies major pollution indicators, it has certain limitations. In contrast, the comprehensive evaluation method accounts for the combined effects of multiple factors, providing a more holistic assessment of water quality and yielding more precise evaluation results (*Ghosh & Bera, 2023*; *Liu et al., 2023*). The overapplication of agricultural fertilizers has led to increased nitrate levels in groundwater (*Zhu et al., 2023*). Groundwater serves as a crucial resource for both drinking and irrigation, significantly impacting human health (*Shi et al., 2024*). *Hou et al. (2023)* utilized a health risk assessment model provided by United States Environmental Protection Agency (USEPA) to assess the human health risks associated with groundwater in the study area and observed that nitrate contamination has become a prominent global groundwater contaminant, posing potential health risks to the population. Prolonged exposure to water with high nitrate levels can impair thyroid function and cause various diseases. Consequently, groundwater contamination has emerged as a critical issue, highlighting the importance of assessing health risks associated with nitrate pollution. Overall, groundwater quality assessment is integral to the rational development and utilization of groundwater resources, and it assesses the status of groundwater quality (*Din, Muhammad & Rehman, 2023*; *Gao et al., 2023b*; *Iqbal et al., 2024*). Each evaluation method has its own advantages and is suited to different needs and scenarios.

Groundwater is the primary source of water for industrial, agricultural, and municipal use in the Baiquan basin (BQB), playing a vital role in social and economic development. However, systematic research on groundwater in the BQB is currently lacking. In this study, the hydrochemical characteristics of groundwater samples from 52 wells, collected during both the dry and wet seasons of 2021, were analyzed using statistical and hydrochemical methods. The water quality of these samples was also assessed. Furthermore, the study evaluated the health risks associated with nitric acid in the groundwater of BQB. The objectives of this study were to: (1) analyze the hydrochemical properties and formation mechanisms of groundwater; (2) evaluate the quality and spatial distribution of groundwater for drinking and irrigation purposes; and (3) assess the non-carcinogenic health risks and spatial distribution of groundwater for different population groups. The findings of this study will provide a scientific basis for the effective management and protection of groundwater resources in the BQB.

## MATERIALS AND METHODS

### Study area
The BQB (Fig. 1) is located in the eastern part of Jinan City, Shandong Province, within a mid-latitude inland region. It experiences a warm temperate continental monsoon climate, characterized by elevated terrain in the south and lower terrain in the north. The southern region predominantly consists of mid-low mountainous terrain and hills, whereas the northern part features a mountain-front alluvial plain. The average annual precipitation in the study area is 671.2 mm, with a spatial and temporal variability in precipitation patterns.

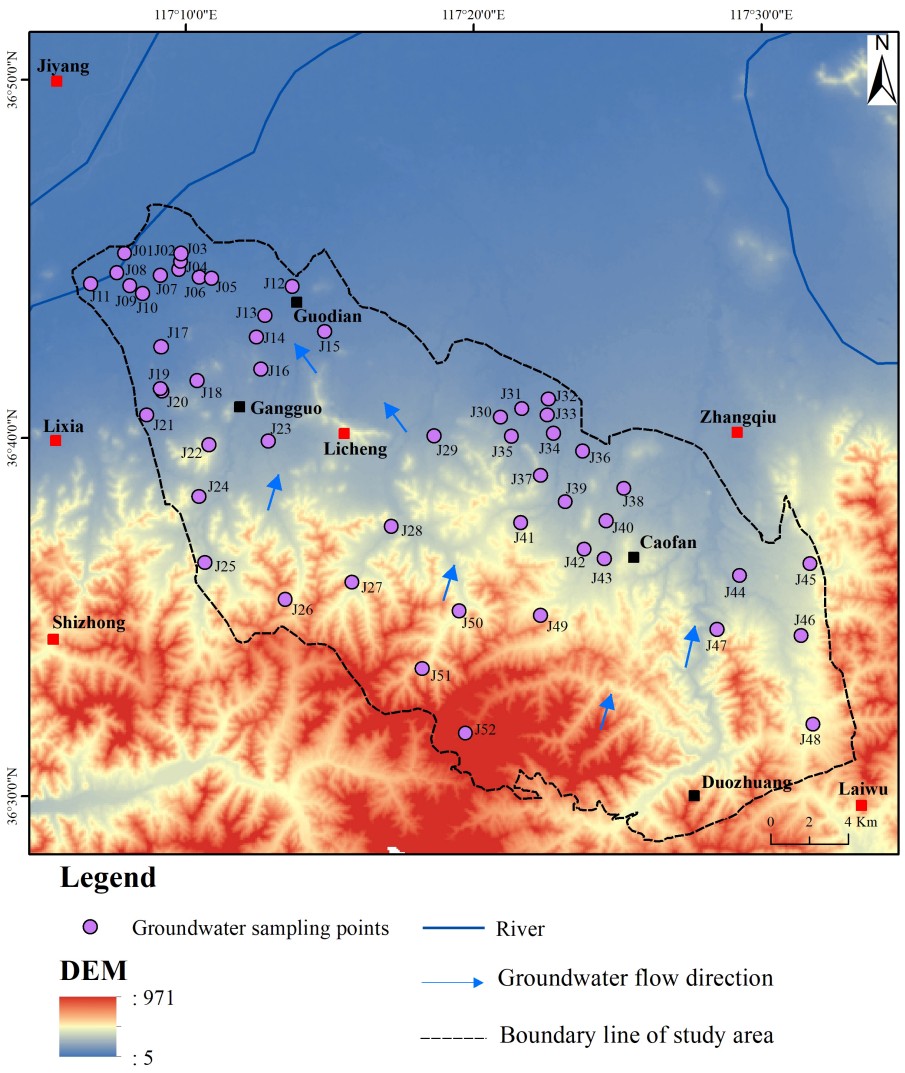

**Figure 1  Geographical context of BQB in northern China and spatial distribution pattern of groundwater monitoring wells.**

Notably, precipitation peaks from June to September, with a spatial decrease observed from the southeastern mountainous regions towards the northwest (*Shu et al., 2021*).

The BQB is located at the northwest margin of the Mount Taishan fault block bulge and features a predominantly north-dipping monoclinic structure primarily composed of Paleozoic strata. The geological activity in this region, especially during the Yanshan Movement, is predominantly characterized by fault activity, with numerous faults and fractures oriented north-northwest (NNW) and north-northeast (NNE), and minimal folding. Stratigraphically, the BQB includes Cambrian, Ordovician, Carboniferous, Permian, and Quaternary strata, although some strata are absent (*Zhang, Xu & Zhang, 2017*).

The BQB exhibits a diverse range of groundwater types, including loose rock pore water, clastic rock pore fissure water, carbonate rock fissure karst water, and bedrock fissure water (Fig. 1). Loose rock pore water is mainly found in the central and northern plain areas of the BQB. The water abundance in these aquifers varies depending on the formation conditions but generally shows moderate richness. Clastic rock pore and fissure water are primarily associated with the Carboniferous Permian sandstone, sand shale, and Tertiary strata. However, the development of pore and fissure water in these layers is limited, resulting in relatively low water abundance. Carbonate fissure karst water is present in regions with thick limestone aquifers, thin limestone layers, and interbedded shale aquifers. The distribution of karst fractures is uneven, leading to significant variations in water abundance.

The recharge, runoff, and discharge processes of the aquifer in the BQB are influenced by a variety of factors, including meteorological, hydrological, stratigraphic, lithologic, and anthropogenic factors such as groundwater extraction and irrigation. Groundwater recharge primarily comes from the infiltration of atmospheric precipitation, supplemented by water from field irrigation. Groundwater runoff generally follows the direction of surface water flow, predominantly moving from southeast to northwest (*Qi et al., 2015*). Natural discharge processes primarily involve lateral runoff and the evaporation of phreatic water. However, the recent increase in groundwater extraction significantly affects groundwater dynamics, with artificial extraction being a major factor. Currently, the primary patterns of groundwater discharge are manual extraction and downstream runoff, with water levels showing considerable variation from south to north.

## Water sampling

In this study, water quality data were collected from 52 wells in BQB both the wet and dry seasons of 2021. For each sample, two 1,000 mL plastic bottles, which were clean, dry, and pre-rinsed with distilled water, were used. Prior to sampling, the bottles were rinsed 2–3 times with the groundwater to be collected. During the sampling process, water was pumped steadily for 10 min to ensure that only fresh groundwater was sampled. The water was gently poured along the sides of the bottle to prevent the introduction of air bubbles or headspace. After filling, the bottles were sealed with a sealing film, labeled with an appropriate identification mark, refrigerated, and then transported (*Sheng et al., 2023*). The water samples were sent to the water chemistry laboratory of Shandong Provincial geo-mineral engineering exploration institute within one week for analysis.

## Water testing

The main chemical components of groundwater were analyzed, including anions ($Cl^-$, $SO_4^{2-}$, $HCO_3^-$, $NO_3^-$, $F^-$) and cations ($K^+$, $Na^+$, $Ca^{2+}$, $Mg^{2+}$), pH, total dissolved solids (TDS), and total hardness (TH). The pH values of the groundwater samples were measured using a pH meter (PHS-3C, Shanghai Yidian Scientific Instruments Co., Ltd., China). Cations, total dissolved solids (TDS), and total hardness (TH) were determined using an inductively coupled plasma emission spectrometer (ICP-OES, Model 7000DV, PerkinElmer, US). $HCO_3^-$ concentrations were assessed through titration, while the
remaining anions were quantified using ion chromatography (ICS-600, Thermo fisher, US). To ensure the accuracy of the water quality data, the charge balance error (CBE) for all samples was calculated using the formula provided in Eq. (1). The results indicate that the CBE values for all samples were within ±5%, demonstrating the reliability of the groundwater quality testing performed in this study (*Li et al., 2018*).

$$\text{CBE} = \frac{\sum cations - \sum anions}{\sum cations + \sum anions} \times 100. \tag{1}$$

## Groundwater quality evaluation
### Drinking water quality
The entropy weight water quality index (EWQI) is an effective method for evaluating water quality, capable of employing multidimensional analysis of multiple parameters to more accurately assess water quality conditions (*Ahmad, Umar & Ahmad, 2024*; *Dashora et al., 2022*; *Liu et al., 2021*). Compared to the traditional water quality index (WQI) method, the EWQI utilizes information entropy to ascertain the weights of each water quality parameter. This approach minimizes the impact of human factors on weight allocation, thereby enhancing the objectivity of the evaluation results. The EWQI method for assessing water quality primarily involves five steps (Fig. 2): establishing an initial water quality matrix, normalizing the data, determining weights using the entropy weight method, establishing grading quantitative standards, and calculating and classifying water quality indices. This study employed the EWQI method to evaluate the quality of groundwater from BQB as drinking water.

### Irrigation water quality
Globally, groundwater not only serves as a source of drinking water but also plays a crucial role in agricultural irrigation (*Kushwah & Singh, 2024*; *Liu et al., 2023*; *Mohammed et al., 2022*). This study assessed the suitability of BQB groundwater for irrigation using two methods: the sodium adsorption ratio (SAR) and the sodium percentage (%Na). The calculation equations for these methods are as follows:

$$\text{SAR} = \frac{Na^+}{\sqrt{\frac{Ca^{2+} + Mg^{2+}}{2}}} \tag{2}$$

$$\%\text{Na} = \frac{Na^+}{Ca^{2+} + Mg^{2+} + Na^+ + K^+}. \tag{3}$$

In Eqs. (2) and (3), the unit for ions is milligram equivalents per liter (meq/L). For SAR values, the categories are as follows: values less than 10, between 10 and 18, between 18 and 26, and greater than 26 correspond to water quality grades of excellent, good, doubtful, and unsuitable, respectively. For Na%, values less than 20, between 20 and 40, between 40 and 60, between 60 and 80, and greater than 80 represent water quality grades of excellent, good, permissible, doubtful, and unsuitable, respectively.

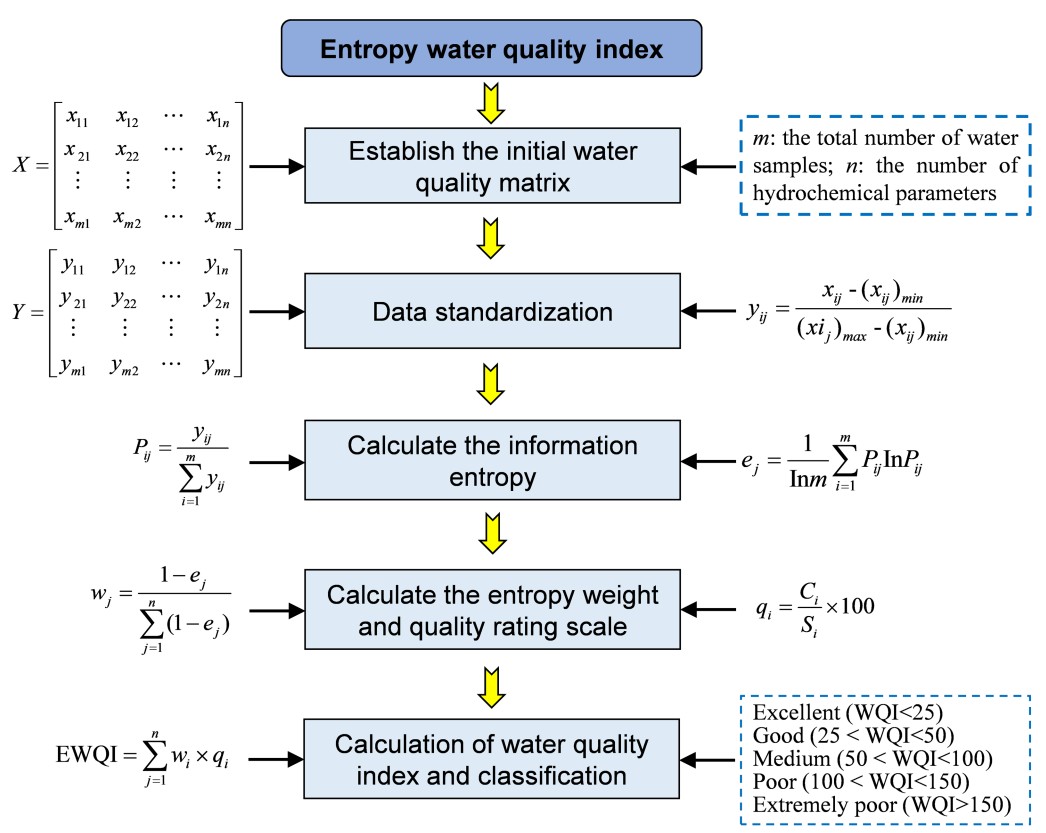

**Figure 2** Computational workflow diagram for determining entropy weight water quality index (EWQI) through integrated physicochemical parameter analysis of surface water samples.

### Health risk assessment

As a source of drinking water, groundwater is also critical to human health risks (*Mao et al., 2023*). The human health risk assessment model of USEPA was adopted to judge whether the local water source was suitable for drinking Eqs. (6) and (7). The calculation formula is as follows:

$$ADD = \frac{C_s \times IR \times EF \times ED}{BW \times AT} \qquad (4)$$

$$HI = \frac{ADD}{RfD}, \qquad (5)$$

when HI < 1, it indicates that the non-carcinogenic risk is an acceptable level of non-carcinogenic risk. An HI > 1 indicates a significant non-carcinogenic risk, and the higher the value, the higher the risk (*Samuel et al., 2024*). The evaluation parameters can be found in Table 1.

**Table 1  Exposure calculation parameter values for HQ value calculation.**

| Parameter | Parameter meaning | Unit | Infant /Child/Adult |
|-----------|-------------------|------|---------------------|
| $C_s$ | Element content | mg/L | Measured value |
| IR | Uptake rate | L/d | 0.5/1/2 |
| EF | Exposure frequency | day/year | 365 |
| ED | Mean exposure time | year | 1/6/6 |
| BW | Average body weight | kg | 10/30/70 |
| AT | Mean exposure time | day | 365*ED |
| RfD | Reference dose | mg/(kg d) | 1.6 |

**Table 2  Statistical results of groundwater hydrochemical composition content.**

| | Wet season | | | | | Dry season | | | | |
|---|---|---|---|---|---|---|---|---|---|---|
| | Max | Min | Mean | SD | CV | Max | Min | Mean | SD | CV |
| $K^+$ (mg/L) | 1.9 | 0.4 | 0.84 | 0.31 | 37% | 1.49 | 0.05 | 0.65 | 0.31 | 48% |
| $Na^+$ (mg/L) | 105 | 5.6 | 19.72 | 20.45 | 104% | 108.1 | 4.09 | 20.29 | 18.08 | 89% |
| $Ca^{2+}$ (mg/L) | 191 | 82.7 | 118.89 | 27.54 | 23% | 317.6 | 63.5 | 129.33 | 46.15 | 36% |
| $Mg^{2+}$ (mg/L) | 41 | 13.1 | 22.66 | 6.63 | 29% | 59.6 | 9.6 | 23.09 | 8.35 | 36% |
| $Cl^-$ (mg/L) | 169 | 13.6 | 40.38 | 33.99 | 84% | 359.6 | 6.8 | 42.81 | 57.63 | 135% |
| $SO_4^{2-}$ (mg/L) | 328 | 41.7 | 97.14 | 55.57 | 57% | 616.9 | 37.2 | 133.83 | 99.13 | 74% |
| $HCO_3^-$ (mg/L) | 434 | 182 | 292.36 | 57.12 | 20% | 380.6 | 178.2 | 280.39 | 40.84 | 15% |
| $NO_3^-$ (mg/L) | 73.2 | 9.8 | 38.31 | 15.42 | 40% | 73.5 | 12.8 | 36.38 | 13.51 | 37% |
| $F^-$ (mg/L) | 0.94 | 0.02 | 0.24 | 0.19 | 79% | 0.85 | 0.1 | 0.19 | 0.12 | 63% |
| pH | 7.44 | 6.9 | 7.22 | 0.16 | 2% | 8.2 | 7.2 | 7.72 | 0.22 | 3% |
| TH (mg/L) | 590 | 270 | 389.82 | 83.51 | 21% | 1,038.3 | 209.2 | 418.02 | 141.99 | 34% |
| TDS (mg/L) | 758 | 362 | 504.08 | 129.55 | 26% | 1,292 | 282 | 536.51 | 198.69 | 37% |

## RESULTS AND DISCUSSION

### General characteristics of groundwater hydrochemistry

The descriptive statistical results for the primary chemical components of BQB's groundwater are presented in Table 2. The average pH values during both the dry and wet seasons were 7.22, with a low coefficient of variation (CV), indicating that BQB groundwater is weakly alkaline and exhibits a relatively stable spatial distribution. TH and TDS are critical parameters for assessing groundwater quality (*Liu et al., 2023*; *Zhang et al., 2022*). During the dry season, TH and TDS in BQB groundwater ranged from 270 mg/L to 590 mg/L and 362 mg/L to 758 mg/L, respectively, with mean values of 389.82 mg/L and 504.08 mg/L (Table 2). In the wet season, the mean values for TH and TDS were 418.02 mg/L and 536.51 mg/L, respectively. Overall, the TH and TDS indicators suggest that the groundwater quality in BQB is generally good, with most water samples classified as hard-fresh water.

On average, $Ca^{2+}$ is the predominant cation in the groundwater of BQB, followed by $Mg^{2+}$. The mean contents of $Ca^{2+}$ and $Mg^{2+}$ during the dry season are 118.89 mg/L and 22.66 mg/L, respectively, whereas during the wet season, they are 129.33 mg/L and

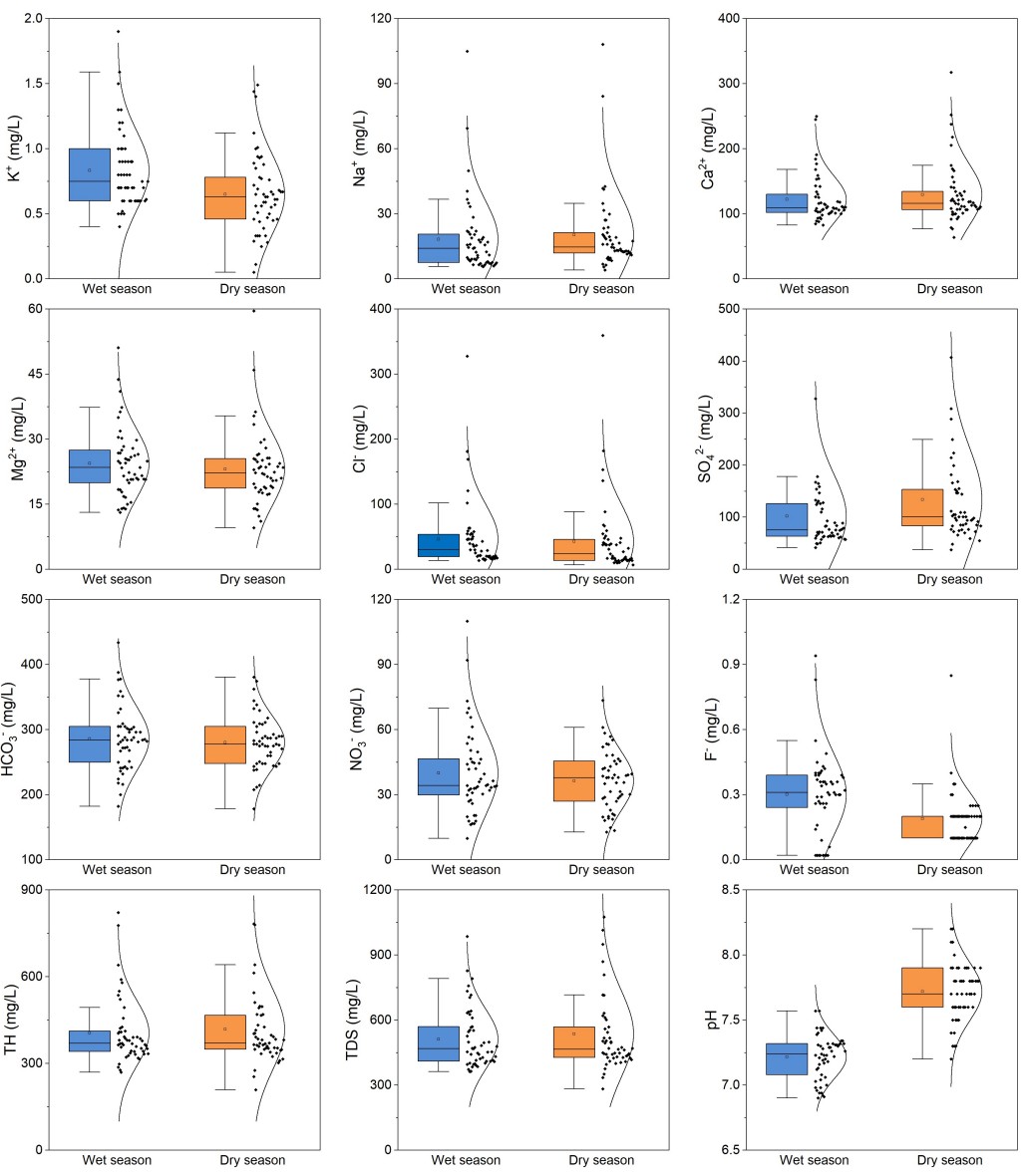

**Figure 3** Statistical distribution analysis using box plots of dominant groundwater chemical constituents (K+, Na+, Ca 2+, Mg 2+, Cl-, $SO_4^{2-}$, HCO₃-, NO₃–, F-, TH, TDS, pH) in BQB.

23.09 mg/L, respectively. $Cl^-$ and $SO_4^{2-}$ are the predominant anions, with average values of 292.36 mg/L and 97.14 mg/L during the dry season, and 280.39 mg/L and 133.83 mg/L during the wet season, respectively. Overall, the composition of groundwater ions in BQB exhibits consistent patterns during both the dry and wet seasons. Cations are dominated by $Ca^{2+} > Mg^{2+} > Na^+ > K^+$, while anions follow the order of $HCO_3^- > SO_4^{2-} > Cl^- > NO_3^- > F^-$ (Fig. 3).

In comparison, the concentrations of $K^+$, $HCO_3^-$, $NO_3^-$, and $F^-$ are higher during the wet season than in the dry season. Conversely, the concentrations of other chemical components are elevated during the dry season, as illustrated in Fig. 3. The nitrate

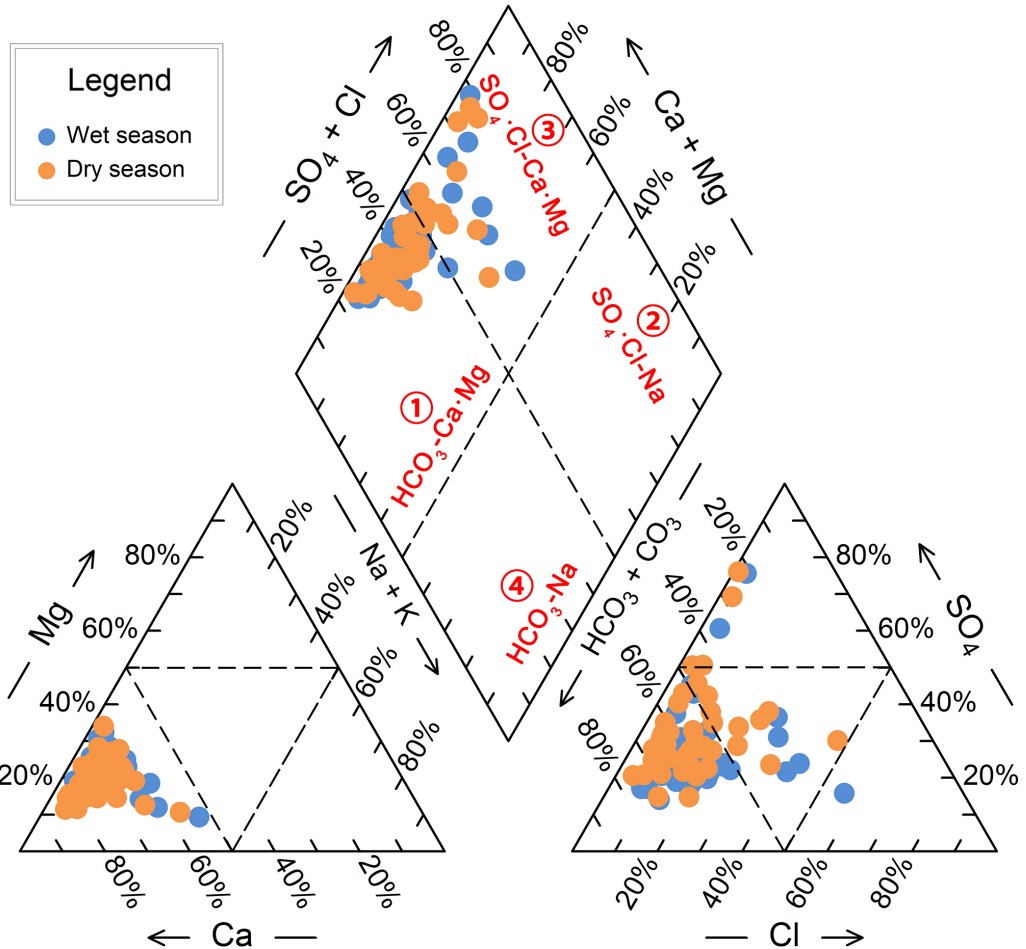

**Figure 4** Hydrochemical classification using Piper trilinear diagram for groundwater samples from BQB in North China.

concentration in BQB's groundwater ranges from 9.8 mg/L to 73.2 mg/L during the dry season and from 12.8 mg/L to 73.5 mg/L during the wet season, with mean values of 38.31 mg/L and 36.38 mg/L, respectively. These values are below the groundwater quality standard limit of 88.57 mg/L. Additionally, the high coefficient of variation for $Na^+$, $Cl^-$, and $F^-$ in BQB's groundwater indicates variability in their concentrations.

## Groundwater hydrochemical types

The Piper trilinear diagram (*Piper, 1944*) is a valuable tool for classifying hydrochemical types, as it is unaffected by human factors and widely used in hydrochemical studies (*Liu et al., 2024*; *Masoud & Abu El-Magd, 2022*). The diagram comprises two triangles and a diamond, with the edges representing the milligram equivalent percentages of anions and cations (Fig. 4). The diamond shape reveals the general chemical characteristics of the

water sample, while the triangles provide information on the relative concentrations of various ions.

When projecting the hydrochemical data of BQB groundwater samples onto the Piper diagram, it is observed that the sample points are predominantly located in the diamond-shaped areas ① and ③. This indicates that the groundwater in BQB can be classified into two hydrochemical types according to the Piper diagram: $HCO_3$-Ca·Mg and $SO_4$·Cl-Ca·Mg (Fig. 4). Specifically, during the wet season, 15.4% of the water samples (eight points) are categorized as the $SO_4$·Cl-Ca·Mg type, while 85.6% are classified as the $HCO_3$-Ca·Mg type. In contrast, during the dry season, 76.9% of the samples (40 points) are identified as the $HCO_3$-Ca·Mg type, with the remaining 23.1% belonging to the $SO_4$·Cl-Ca·Mg type. The Baotu spring area, adjacent to BQB, is also a karst spring region. The hydrochemical type in the Baotu spring area is similar to that of BQB, primarily characterized by the $HCO_3$-Ca type and $HCO_3$·$SO_4$-Ca. Additionally, the Cl·$SO_4$-Ca·Mg and $HCO_3$-Ca·Mg types are also present (*Wang et al., 2024a*; *Wang et al., 2024b*).

## Controlling factors of groundwater hydrochemistry
### *Gibbs graphical model*
Groundwater hydrochemical characteristics result from a combination of natural processes and human influences (*Alvarez et al., 2020*; *Gao et al., 2023a*; *Liu et al., 2024*; *Zhang et al., 2022*). *Gibbs (1970)* developed a graphical model for analyzing surface water hydrochemistry, which has been extensively applied in groundwater studies (*Liu et al., 2024*; *Masoud & Abu El-Magd, 2022*). The Gibbs model (Fig. 5) utilizes a semi-logarithmic coordinate system divided into three regions, representing key factors affecting hydrochemical characteristics: rock weathering, evaporation, and atmospheric precipitation (*Gibbs, 1970*). However, the Gibbs model has limitations in assessing the impact of human activities on water chemistry. Projection of the BQB groundwater sample data onto the Gibbs model (Fig. 5) shows that the majority of the data points fall within the central region, corresponding to the rock weathering control zone. This distribution indicates that the hydrochemical composition of BQB groundwater is predominantly influenced by rock weathering processes.

### *Ion ratios analysis*
Ion ratio end-member plots are useful for identifying the types of rock weathering sources influencing hydrochemical composition (*Liu et al., 2024*; *Masoud & Abu El-Magd, 2022*; *Ren et al., 2020*). According to *Gaillardet et al. (1999)*, the end-member diagram is divided into three areas: evaporite rocks in the bottom left, silicate rocks in the middle left, and carbonate rocks in the top right. When the groundwater hydrochemical data of BQB are projected onto the end-member plot (Fig. 6), the sample points are predominantly distributed between the carbonate rock and silicate rock regions. This distribution indicates that the water chemistry of the BQB groundwater is predominantly influenced by the weathering of carbonate and silicate rocks.

In theory, if $Na^+$ and $Cl^-$ in groundwater primarily originate from the dissolution of halite (NaCl), the $Na^+$/$Cl^-$ ratio should be approximately 1 (*Li et al., 2018*; *Ren et al.,*

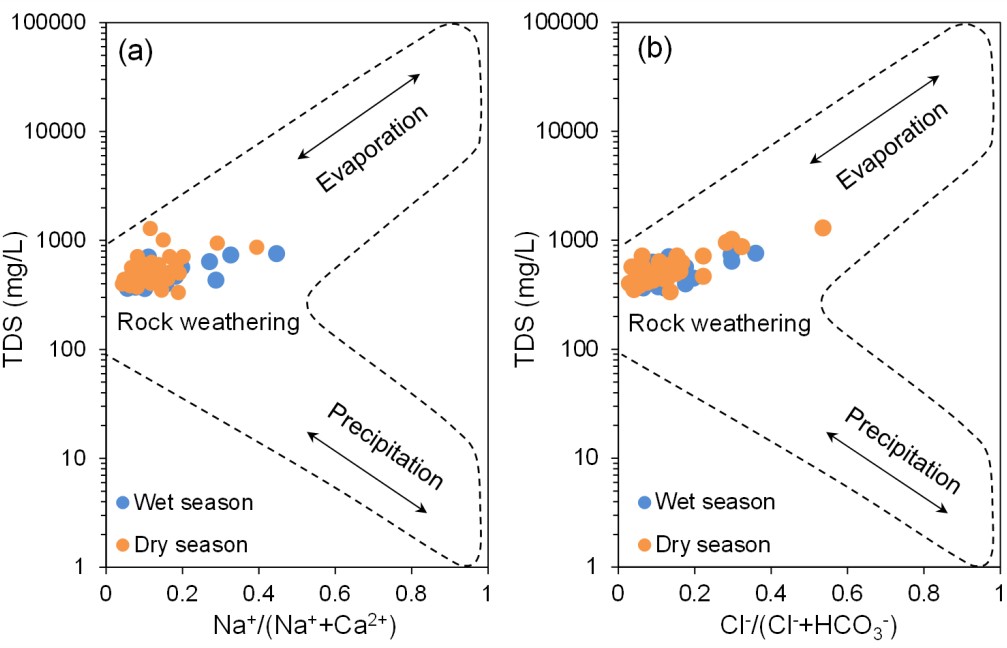

**Figure 5** Hydrochemical evolution mechanisms revealed by Gibbs binary diagrams for groundwater samples in BQB.

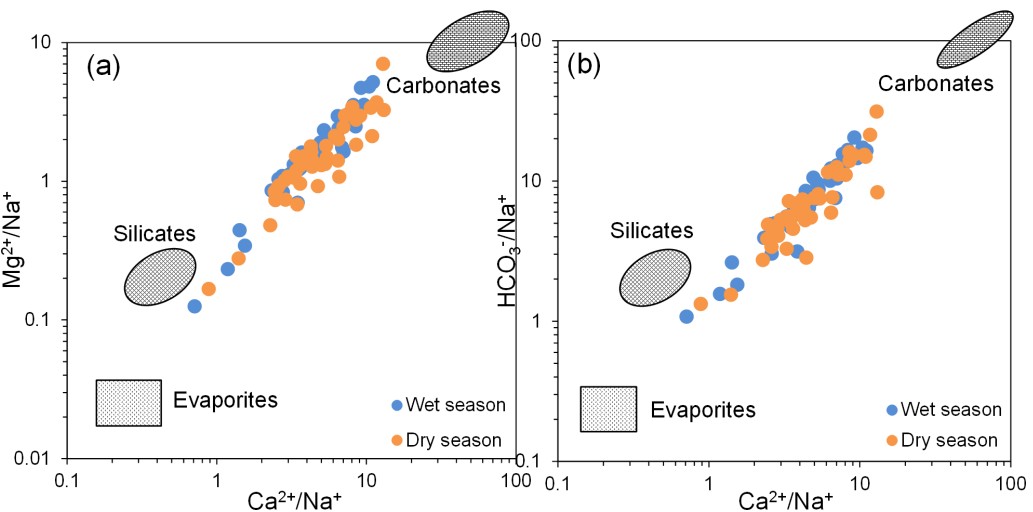

**Figure 6** End member diagram of groundwater hydrochemical ion ratios. End member diagram of groundwater hydrochemical ion ratios, (A) $Mg^{2+}/Na^+$ *vs.* $Ca^{2+}/Na^+$, (B) $HCO_3^-/Na^+$ *vs.* $Ca^{2+}/Na^+$.

*2020*). As illustrated in Fig. 7A, the majority of groundwater sampling sites at BQB are distributed on either side of the $Na^+/Cl^- = 1$ line, with several sites in close proximity to this line. This distribution suggests that rock salt dissolution is not the sole source of $Na^+$ and $Cl^-$. Water samples located above the 1:1 line exhibit $Cl^-$ concentrations

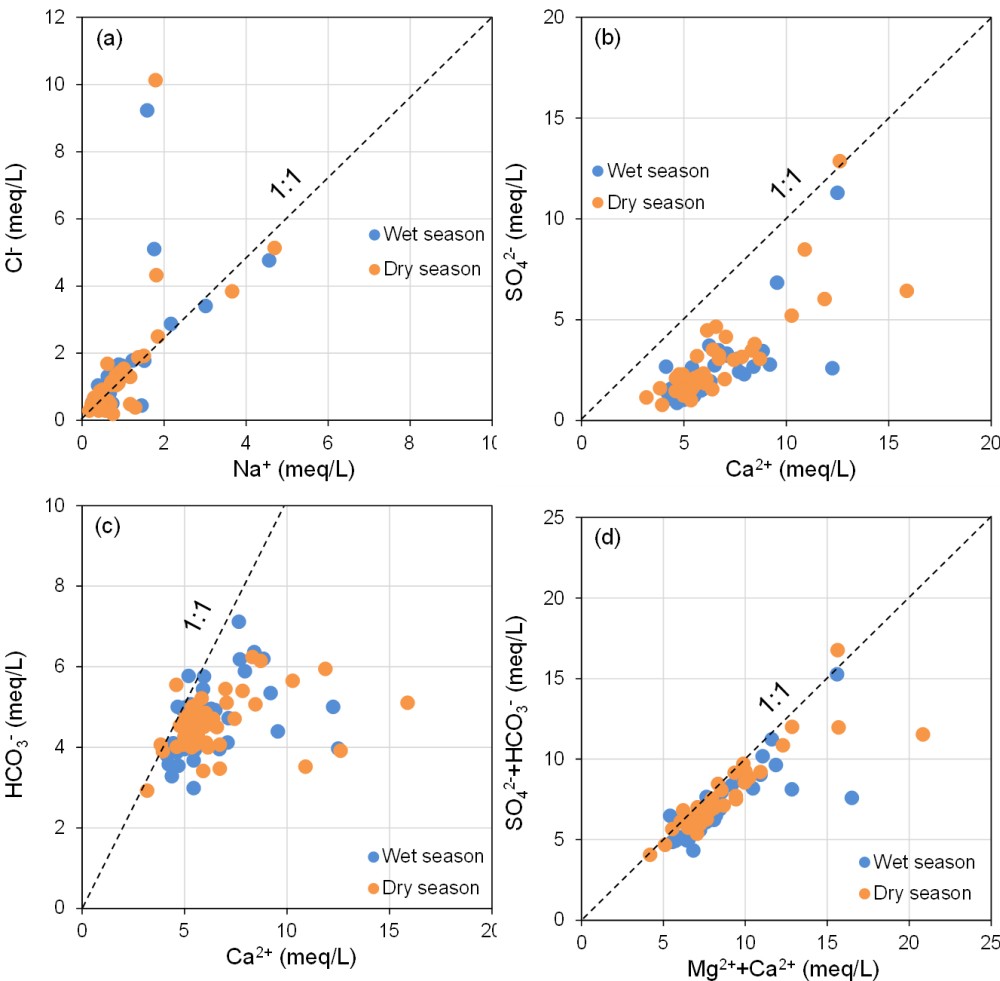

**Figure 7 Main ion ratio diagram of BQB groundwater, (A) Cl⁻ vs. Na⁺, (B) SO₄²⁻ vs. Ca²⁺, (C) HCO₃⁻ vs. Ca²⁺, (D) (SO₄²⁻ + HCO₃⁻) vs. (Mg²⁺ + Ca²⁺).**

that exceed Na⁺ levels, indicating potential anthropogenic influences (*Liu et al., 2021*). Conversely, water samples below the 1:1 line show higher Na⁺ concentrations, likely due to the weathering and dissolution of silicate minerals, which can result in excess Na⁺. Additionally, cation-exchange processes within the aquifer may also impact Na⁺ concentrations (*Zhang et al., 2022*). Figure 7B shows that BQB groundwater samples are mainly located to the right of the $Ca^{2+}/SO_4^{2-} = 1$ line. This distribution implies that $SO_4^{2-}$ is not sufficient to balance the $Ca^{2+}$ concentration, suggesting that gypsum dissolution is not the sole source of $Ca^{2+}$ and $SO_4^{2-}$ in the groundwater. Given that the study area's hydrogeology is primarily composed of carbonate rocks, the excess $Ca^{2+}$ is likely derived from the dissolution of carbonate minerals (*Qi et al., 2015*).

The relationship between $Ca^{2+}$ and $HCO_3^-$ can be utilized to determine the contribution of carbonate mineral dissolution to the concentrations of $Ca^{2+}$ and $HCO_3^-$ in groundwater (*Li et al., 2018*; *Zhang et al., 2022*). As depicted in Fig. 7C, the majority of the groundwater

samples from BQB are distributed predominantly on the right side of the $Ca^{2+}/HCO_3^- = 1$ line, with some samples located near this line. This distribution suggests that carbonate mineral (*e.g.*, calcite) (*Yang et al., 2016*) dissolution contributes to the levels of $Ca^{2+}$ and $HCO_3^-$, although $Ca^{2+}$ also originate from other sources. If the dissolution of carbonate and sulfate minerals is the dominant hydrogeochemical reaction in a groundwater system, the molar ratio $(Ca^{2+}+Mg^{2+})/(HCO_3^-+SO_4^{2-}) = 1$ is theoretically approximated to 1. Ratios greater than 1 suggest that silicate weathering predominantly influences the groundwater chemistry. Conversely, ratios less than 1 indicate that the hydrochemistry of groundwater is primarily driven by the weathering of carbonate rocks (*Liu et al., 2024*). Figure 7D illustrates that the water sample points are primarily situated below the $(Ca^{2+}+Mg^{2+})/(HCO_3^-+SO_4^{2-}) = 1$ line, indicating that the main hydrogeochemical process affecting the groundwater in BQB is silicate rocks weathering, with some influence from carbonate rocks weathering as well. The ionic components of the Baotu spring are primarily influenced by the dissolution of rock salt, carbonate, and sulfate minerals (*Wang et al., 2024a*; *Wang et al., 2024b*).

Cation exchange, alongside mineral dissolution and precipitation, frequently contributes to the regional hydrochemical composition of groundwater (*Gao et al., 2023b*; *Hu et al., 2024*; *Ige, Adewoye & Obasaju, 2021*; *Ren et al., 2020*). This process can be identified and quantified through the relationship between $(Na^++K^+-Cl^-)$ and $(Ca^{2+}+Mg^{2+}-HCO_3^--SO_4^{2-})$. If cation exchange is the primary process influencing the hydrogeochemical properties of groundwater, the relationship between the relevant variables should be linear, with a slope approximating $-1$ (*Zhang et al., 2024*). The application of the chlor-alkali index (CAI), as outlined in Eqs. (6) and (7), enables the determination of the direction of cation exchange. If both CAI-1 and CAI-2 are negative, cation exchange is occurring within the groundwater system. Conversely, positive values for both indices indicate the occurrence of reverse cation exchange (*Hu et al., 2024*; *Liu et al., 2024*).

$$CAI-1 = \frac{Cl^- - (Na^+ + K^+)}{Cl^-} \tag{6}$$

$$CAI-2 = \frac{Cl^- - (Na^+ + K^+)}{HCO_3^- + SO_4^{2-} + CO_3^{2-} + NO_3^-}. \tag{7}$$

As illustrated in Fig. 8A, the distribution of water sample points near the $(Na^++K^+-Cl^-)/(Ca^{2+}+Mg^{2+}-HCO_3^--SO_4^{2-}) = -1$ line suggests that cation exchange significantly influences the hydrochemical characteristics of groundwater in BQB. Furthermore, Fig. 8B indicates that the majority of groundwater sample points fall within the area where CAI > 0, revealing that reverse cation exchange is predominant in the BQB groundwater environment, characterized by the exchange of $Na^+$ in groundwater with $Ca^{2+}$ in the aquifer medium.

The impact of anthropogenic activities on groundwater environments is increasingly evident, particularly in regions with advanced industrial and agricultural activities. Human activities are a significant factor influencing the hydrochemical composition of groundwater (*Abdulsalam et al., 2022*; *Adjëï Kouacou et al., 2024*). Nitrate ($NO_3^-$) is the primary anthropogenic source in groundwater and can serve as an indicator of the intensity

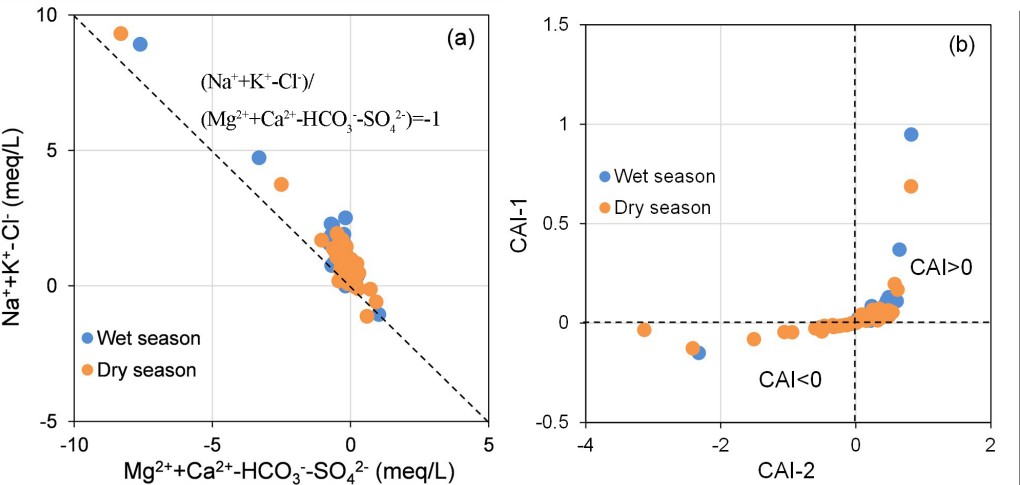

**Figure 8** Hydrogeochemical diagnostic plots of $[(Na^++K^+)-Cl^-]$ *vs.* $[(Ca^{2+}+Mg^{2+})-(HCO_3^-+SO_4^{2-})]$ (A) and chloro-alkaline indices (CAI-1 *vs.* CAI-2) (B) for groundwater samples.

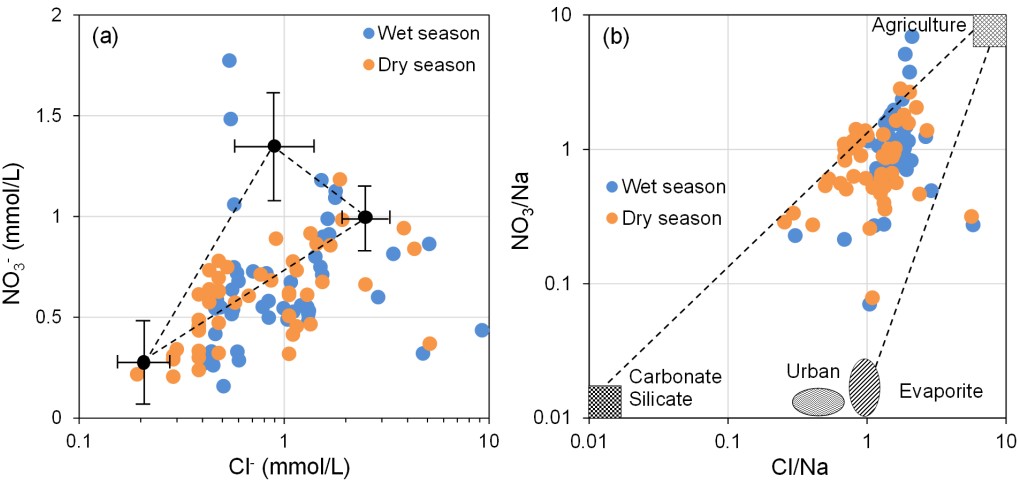

**Figure 9** Relationship between $NO_3^-$ *vs.* $Cl^-$ (A), variations in Cl/Na with $NO_3$/Na (B), used to identify the source of nitrate.

of human activities to some extent (*Garrard et al., 2023*; *Liu et al., 2021*). $NO_3^-$ and $Cl^-$ are commonly used to identify and analyze disturbances in groundwater chemical features due to anthropogenic activities (*Hua et al., 2020*; *Liu et al., 2021*).

As depicted in Fig. 9A, the majority of groundwater sampling sites in the BQB are situated in the lower left corner, within the fertilizer control area. Furthermore, Fig. 9B shows that water sample points are predominantly clustered near the agricultural impact area in the upper-right corner, indicating that the $NO_3^-$ in BQB's groundwater is largely attributed to the use of chemical fertilizers in agricultural activities.

## Groundwater quality evaluation
### Drinking water quality

Evaluating the water quality of groundwater, which serves as a primary source of water for industry and agriculture, is crucial for the rational development and utilization of regional groundwater resources (*Ahmad, Umar & Ahmad, 2024*; *Ghosh & Bera, 2023*; *Kammoun, Abidi & Zairi, 2022*; *Kushwah & Singh, 2024*). In this study, EWQI was employed to assess groundwater quality in the study area. Based on the calculated EWQI values, groundwater quality was categorized into five grades, ranging from ''very poor'' to ''very good.'' Generally, EWQI values exceeding 100 indicate that the water quality is unsuitable for drinking purposes (*Liu et al., 2023*). The assessment of groundwater drinking water quality using the EWQI for BQB revealed that EWQI values during the wet season ranged from 27.01 to 78.75, with an average of 40.41. In contrast, during the dry season, EWQI values ranged from 25.12 to 93.46, with an average of 43.62. Overall, BQB exhibits good groundwater quality and is considered a suitable source of drinking water.

The water quality assessment results indicated that the water quality during both the dry and wet seasons was essentially consistent, falling into two primary categories: good and moderate, with good being the predominant classification. During the dry season, 42 water samples (80.77%) were classified as good, and 10 (19.23%) as medium. Similarly, during the wet season, 41 water samples (78.85%) were classified as good, and 11 (21.15%) as medium. Figure 10 illustrates the spatial distribution of EWQI values for groundwater quality in BQB. The results indicate that the spatial distribution of groundwater quality assessment results is generally consistent between the dry and wet seasons, with higher EWQI values observed in the northwestern and southern parts of the BQB. This pattern may be attributed to the higher frequency of human activities in these regions, such as industrial and agricultural pollution, as well as over-exploitation of groundwater (*Kong et al., 2015*). Additionally, the northern part of the BQB exhibited lower EWQI values during the rainy season compared to the dry season, likely due to elevated TDS and fluoride concentrations resulting from seasonal evapotranspiration and concentration effects in the dry zone. Based on these findings, the government can prioritize targeted water quality control and improvement measures in the study area.

### Irrigation water quality

High-quality irrigation water provides essential nutrients and moisture for plant growth and development. However, substandard irrigation water quality can result in issues such as soil salinization, soil acidification, and plant toxicity (*Bhatnagar & Thakral, 2024*; *Kushwah & Singh, 2024*; *Liu et al., 2023*). To assess the suitability of BQB's groundwater for agricultural irrigation, this study used SAR and Na% as key indicators.

The results indicate that the SAR values of BQB groundwater range from 0.13 to 2.34 during the dry season and from 0.09 to 2.11 during the wet season, with mean values of 0.39 and 0.40, respectively. All water samples fall into the excellent category (SAR < 10), suggesting that BQB's groundwater is suitable for agricultural irrigation. Additionally, the Na% values range from 2.97 to 37.34 during the dry season and from 2.44 to 32.19 during the wet season, with mean values of 8.35 and 8.99. During the dry season, all water samples

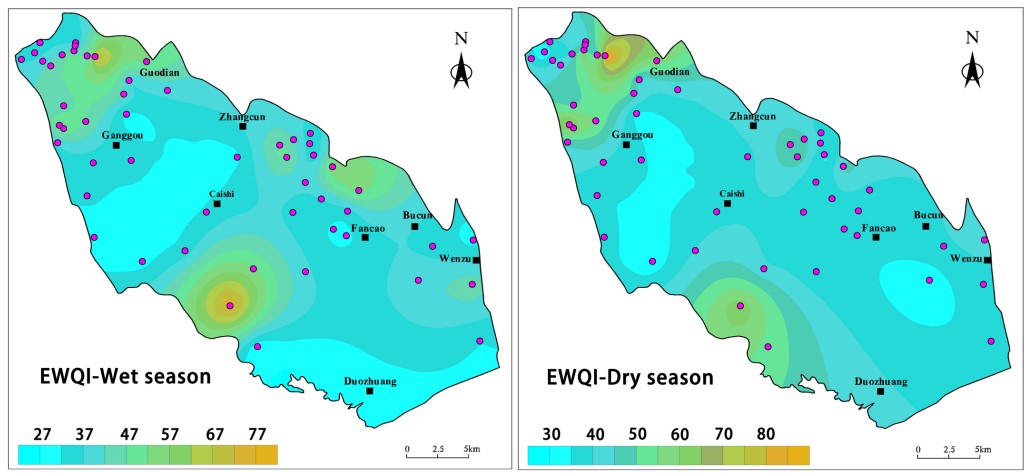

**Figure 10** Geospatial variability mapping of EWQI for groundwater in BQB, (A) wet season, (B) dry season.

except for four, which are classified as good (20 < Na% < 40), are categorized as excellent (Na% < 20). During the wet season, all water samples except for two, which are classified as good, also fall into the excellent category. Thus, based on Na%, BQB's groundwater also exhibits excellent water quality for irrigation purposes.

The United States Salinity Laboratory (USSL) irrigation water quality classification diagram (*Liu et al., 2023*) reveals that the groundwater samples are primarily clustered in two zones, namely $C_2S_1$ and $C_3S_1$ (Fig. 11A). This distribution pattern suggests that BQB's groundwater qualifies as an ideal source for agricultural irrigation. Furthermore, according to the Wilcox irrigation water quality classification diagram (*Akakuru et al., 2022*; *Liu et al., 2023*), the groundwater points are predominantly located in regions ① and ② (Fig. 11B). This distribution indicates the excellent irrigation water quality of BQB's groundwater. In summary, BQB's groundwater stands out as a superior source for agricultural irrigation water supply.

### Health risk assessment

The extensive use of inorganic nitrogen fertilizers in agriculture has significantly increased nitrate concentrations in water. Nitrate is highly soluble and stable in water. Human exposure to nitrates occurs primarily through drinking water and food, with nitrates being readily absorbed and excreted by organisms, playing an essential role in the nitrogen cycle of river basins. The health risk assessment model developed by the *USEPA (2001)* was employed to evaluate nitrate-related health risks in drinking water for infants, children, and adults, using the hazard quotient (HQ) as the characterization metric.

The calculation results demonstrate that during the wet season, the health HQ values for nitrate-related health risks in groundwater exhibit a range of 0.31 to 3.44 for infants, 0.20 to 2.29 for children, and 0.18 to 1.96 for adults, with respective mean values of 1.25, 0.83, and 0.71. Conversely, during the dry season, these ranges are 0.40 to 2.30 for infants,

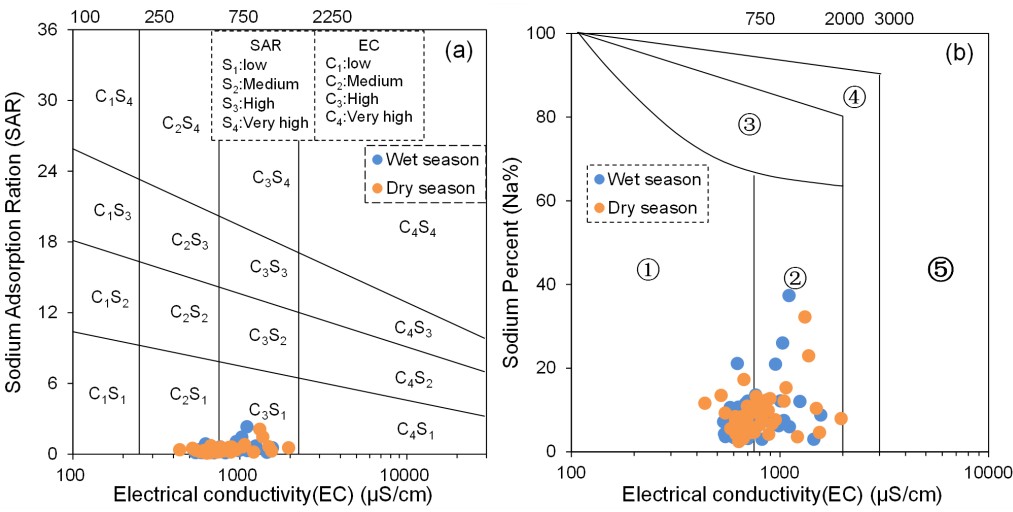

**Figure 11   USSL diagram (SAR *vs* EC) (A) and Wilcox diagram (Na% *vs*. EC) (B) of irrigation water quality classification.**

0.27 to 1.53 for children, and 0.23 to 1.31 for adults, yielding average HQ values of 1.14, 0.76, and 0.65, respectively. Overall, the health risk associated with nitrate in groundwater appears to be higher during the wet season, possibly due to increased precipitation in the wet season facilitating the leaching of nitrates from soil into groundwater.

A spatial distribution map of nitrate health risks in the groundwater of BQB was generated using GIS software, as depicted in Fig. 12. The figure shows that the non-carcinogenic risk is mainly concentrated in the north-western and south-eastern parts of the study area, while higher EWQI values are observed in the north-western and southern parts of the BQB, which may be due to the high level of anthropogenic activities in these areas, such as agricultural fertilisers, manure, and domestic wastewater, which can degrade water quality (*Peng, 2023*). Overall, HQ values were slightly higher in infants compared to children and adults. In the dry season, 65% of the sampling sites posed a significant non-cancer risk to infants, while in the wet season, this proportion was 56%. Comparing the health risks associated with BQB groundwater to those of the Jinan spring area, it was observed that the nitrate non-carcinogenic health risk index exhibited an increasing trend with decreasing receptor age in both regions. However, the health risk index was comparatively higher in the BQB than in the Jinan spring area (*Dou et al., 2022*). These findings underscore the necessity for further investigation and mitigation strategies.

By comparing the EWQI water quality evaluation with the health risk assessment, it was observed that water samples classified as good or moderate still exhibited health risks exceeding permissible limits. This discrepancy was attributed to the use of the USEPA human health risk assessment model, which employs nitrate limits that differ from those utilized in the EWQI. It is recommended that a unified health risk assessment system be established to enhance the accuracy of groundwater health risk assessments. Additionally, protecting groundwater from pollution is essential. This involves reducing the overuse of

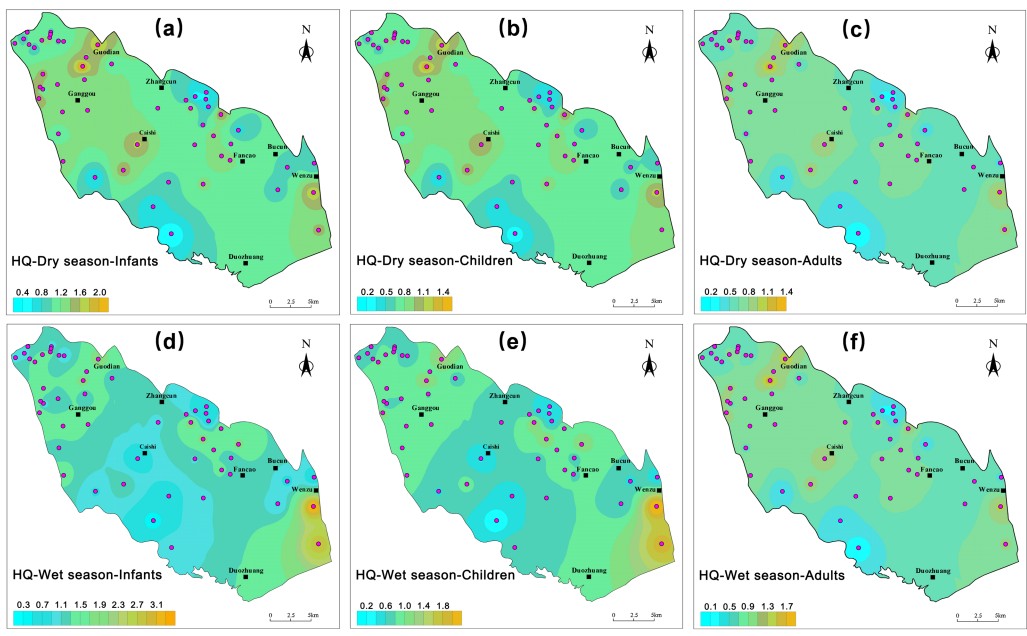

**Figure 12** Spatial distribution map of non-carcinogenic health risks of nitrate in groundwater, including infants, children, adults.

fertilizers and pesticides in agriculture. Regular assessments of groundwater quality are conducted, particularly for contaminants that pose significant risks to children's health.

## CONCLUSIONS

Groundwater serves as a crucial source of industrial and agricultural water for BQB. This study analyzed hydrochemical data from both dry and wet seasons to characterize and identify the factors influencing groundwater chemistry and to evaluate its quality. The key findings are as follows:

(1) Groundwater in BQB is weakly alkaline and primarily classified as hard-fresh water. The concentration order of cations in groundwater is $Ca^{2+} > Mg^{2+} > Na^+ > K^+$, and for anions, it is $HCO_3^- > SO_4^{2-} > Cl^- > NO_3^- > F^-$. Notably, concentrations of $K^+$, $HCO_3^-$, $NO_3^-$, and $F^-$ are higher during the dry season compared to the wet season. Piper diagram analysis reveals that the groundwater's hydrochemical type is predominantly $HCO_3$-Ca·Mg type, followed by $SO_4$·Cl-Ca·Mg type.

(2) The hydrochemical composition of the groundwater in the BQB is primarily influenced by the weathering of silicate and carbonate rocks. Directed cation exchange also impacts the concentrations of $Ca^{2+}$ and $Na^+$ in the groundwater. The presence of $NO_3^-$ is mainly attributed to the use of fertilizers in agricultural activities.

(3) Drinking water quality assessments categorize groundwater into two main classes: good and moderate, with the majority being classified as good. The spatial distribution of the EWQI for groundwater exhibits a general consistency between the dry and wet seasons. However, the northern part of the BQB demonstrates lower EWQI values during the wet

season as compared to the dry season. Regarding irrigated agriculture, SAR and Na% values, along with irrigation water quality classification, confirm that the groundwater in the BQB is an excellent resource for agricultural use in the region.

(4) The results of the health risk assessment reveal that the mean HQ scores for infants, children, and adults during the wet season are 1.25, 0.83, and 0.71, respectively. Conversely, during the dry season, HQ values for infants, children, and adults decreased to 1.14, 0.76, and 0.65, respectively. Consequently, the health risk associated with nitrate contamination in the groundwater of BQB is notably elevated during the wet season. Additionally, the high-risk zones for nitrate contamination in BQB's groundwater are primarily concentrated in the northwest and southeast regions.

(5) Based on the results, local governments can utilize water quality distributions to identify priority areas for groundwater pollution control, implement stringent pollution control measures, and develop comprehensive plans for groundwater protection and utilization. Additionally, the health risk assessment highlights the need for local governments to strictly control groundwater pollution and implement protective measures for adolescents and children. This study analyzed data from a single year; future efforts will include dynamic monitoring in the study area to gain a deeper understanding of the groundwater quality situation.

## ACKNOWLEDGEMENTS

We would like to express our sincere thanks to the editors and reviewers for their very helpful comments for the paper.

### Funding

This study was financially supported by National Natural Science Foundation of China (No. 42272288) and the Shandong Provincial Geological Exploration Project ((Lu Kan Zi (2023) No. 4). The funders had no role in study design, data collection and analysis, decision to publish, or preparation of the manuscript.

### Grant Disclosures

The following grant information was disclosed by the authors:
National Natural Science Foundation of China: 42272288.
Shandong Provincial Geological Exploration Project: No. 4.

### Competing Interests

Zhe Zhang and Mingguo Wang are employed by Shandong Hydrogeology Engineering Geologyand Environment Geology Corporation.

### Author Contributions

- Bo Li conceived and designed the experiments, performed the experiments, authored or reviewed drafts of the article, and approved the final draft.

- Di Wu conceived and designed the experiments, performed the experiments, prepared figures and/or tables, authored or reviewed drafts of the article, and approved the final draft.
- Dalu Yu conceived and designed the experiments, performed the experiments, authored or reviewed drafts of the article, and approved the final draft.
- Changsuo Li performed the experiments, authored or reviewed drafts of the article, and approved the final draft.
- Jinxiao Wang performed the experiments, prepared figures and/or tables, and approved the final draft.
- Liting Xing performed the experiments, analyzed the data, prepared figures and/or tables, and approved the final draft.
- Shuai Gao performed the experiments, analyzed the data, prepared figures and/or tables, and approved the final draft.
- Zhe Zhang performed the experiments, analyzed the data, prepared figures and/or tables, and approved the final draft.
- Mingguo Wang performed the experiments, analyzed the data, prepared figures and/or tables, and approved the final draft.
- Yuwei Wang performed the experiments, analyzed the data, authored or reviewed drafts of the article, and approved the final draft.

## Data Availability

The raw data is available in the Supplementary File.

## Supplemental Information

Supplemental information for this article can be found online at http://dx.doi.org/10.7717/peerj.19233#supplemental-information.

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
