# Peer review of "Comprehensive analysis of groundwater hydrochemistry and nitrate health risks in the Baiquan basin, Northern China"

_PeerJ, doi:10.7717/peerj.19233_

## Round 0.1 · original submission · Major Revisions

· Academic Editor

Major Revisions

Based on the comments from four anonymous reviewers, your article required major revision. Particularly, more comprehensive literature review, more detailed description on methodology, and deeper discussion with comparison with previous studies are needed.

Reviewer 1 ·

Basic reporting

Dear Editor,

Thank you for inviting me to review this manuscript. In the manuscript I am reviewing, investigates the hydrochemical properties and controlling factors of groundwater in the Baiquan basin by analyzing water quality data collected during both dry and wet periods. In my opinion this study is of a certain significance, and its findings have a guiding value for the management and development of groundwater resources in the study area. Overall, the manuscript is well-structured with aesthetically pleasing diagrams, suggesting that the authors had a systematic exercise in scientific research. Therefore, I think this manuscript can be accepted by the journal after minor revision.

I propose the following revisions to the manuscript.

1) The introduction did not highlight the methods used in the manuscript. The authors should improve this section in the introduction.
2) Please carefully check the format of the subscripts for ions in the manuscript.
3) 'Study area' must be moved into the material and methods section.
4) All reagents (purity, company, city, country) and all instruments (company, city, and country) must be included.
5) In lines 169-174, the authors should add relevant references.
6) The authors should add relevant references in lines 175-178.
7) The quality of the discussion section can be improved and avoid duplicating the introduction..
8) The conclusions of your paper are especially important for this. Therefore, please try to sharpen this further.

Experimental design

no comment

Validity of the findings

no comment

Additional comments

no comment

Reviewer 2 ·

Basic reporting

.

Experimental design

.

Validity of the findings

.

Additional comments

The manuscript is a generalized paper. I suggest the following to improve the manuscript.

1) The scope and purpose of the study should be emphasized in the introduction section.
2) Line 125: Specify the year of the sampling
3) Line 181: CV is expressed in %
4) Lines 225 and 226: How did the authors make the following statement?
5) Gibbs model has limitations in assessing the impact of human activities on water chemistry.

Reviewer 3 ·

Basic reporting

Introduction section is lack of literature collection, Discuss the studies published in recent years

Experimental design

Discuss the methodology of sampling techniques and testing methods with detailed references

Validity of the findings

Result and discussion section need to revise and compare the present studies with recent studies

Additional comments

Abstract should be more informative and should add key findings

Revise conclusion based on the corrections

Reviewer 4 ·

Basic reporting

The manuscript is generally clear and well-organized. However, some sections could benefit from more concise language to improve readability. Somes suggestions have been made in the document.

The study is well-contextualized within the existing literature, with appropriate references to previous research. Figures and tables are well-designed but could benefit from informative captions. The references are relevant and up to date, supporting the study’s background and rationale effectively.

Experimental design

Additional information on the sampling process could enhance transparency.

The methods used in this study are not described in detail, lack justification, and fail to address any limitations.

Comments are provided in the document where necessary.

Validity of the findings

The authors should focus on evaluating the meaning and significance of the results. They need to clarify the findings and thoroughly discuss their relationship to the geology and activities of the study area that affect its quality. Currently, the justification for the findings is shallow, speculative, and lacks sufficient scientific evidence. Additional effort is required to strengthen the discussion. The authors are advised to critically revisit this section.

Additional comments

The discussions section could be deeper, particularly regarding the implications of the findings. The authors should deepen the discussion on the implications of the findings for groundwater management and public health. Address potential limitations of the study and suggest areas for future research.

Annotated reviews are not available for download in order to protect the identity of reviewers who chose to remain anonymous.

---

## Round 0.2 · Minor Revisions

· Academic Editor

Minor Revisions

Although we have not received the comments from another reviewer, we want make a decision of Minor Revision based on the comments from reviewer 4 to accelerate the review process. If there will be further comments from the other reviewer before your manuscript being accepted, we will feed back to you in time.

Reviewer 4 ·

Basic reporting

Here are a few comments to improve the paper:

Line 108 -141: In the study area section, none of the statements were referenced. Could you please clarify whether the information presented is original or based on existing research? If the latter, kindly provide relevant citations to support the claims.

Line 154: institute should start with a capital letter (Geo-mineral Engineering Exploration institute).

Line 294 – 295 The sentence “The excess Ca2+ might be derived from the dissolution of carbonate minerals or Ca-containing silicate minerals”. Are these minerals in the study area? You need to authenticate this sentence by stating the typical carbonate and silicate minerals in the study area that are contributing to these ions in the area. Provide appropriate references of works on the geology of the area to support the claim. Again, in Line 301, is calcite found in the study area? What is the justification for it? Same comments for Lines 309 – 312.

Line 325 – 326 As illustrated in Figure 8a, the distribution of water sample points near the (Na++K+-Cl⁻)/(Ca2++Mg2+-HCO3⁻-SO42⁻)=⁻) = 1. Is it -1 or 1? Please show the equation on the figure.


Line 204-205: Could there be a reason for higher K+, HCO3, NO3-, and F- during dry season than in the wet season?

Line 351: check the sentence: is it rational development or you mean national development?

Line 354 – 356: What guideline values did you use for comparison? Please state the guideline values and support it with appropriate reference.

Line 360 – 367 What is the reason for the disparity in the dry and wet seasons? Again, for Fig. 10, what is the reason for the distribution observed across the study area? It will be good to explain beyond reasonable doubt what could be causing the high values in certain parts and low in other areas of the study are? Ensure you support your claims with relevant literature.

Line 405 – 407: Relate this statement with activities and nature of the geology of the study area and support it with relevant literature of similar situations elsewhere. Line 409 – 411 Give reasons for these occurrences observed. For instance, why is the non-carcinogenic risks predominantly concentrated in the northwest and southeast regions of the study area and not the other parts of the study area? Give reasons for every claim you make and support it with relevant literature.

Line 411 – 413: The sentences here are a bit confusing. Wet season elevated compared to dry season. And later mentioned 65% in dry season and 56% in wet season. It is contradictory.

Line 430 - 432: Conclusion: The 3rd and 4th findings were not well captured in the introduction, especially at the objectives.
What is the future direction of this research? It will be good to state some recommend at the Conclusion Section too.

Lines 226 -228 and 436 – 437: concentrations of K⁺, HCO₃⁻, NO₃⁻, and F⁻ are higher during the dry season compared to the wet season. What could be the reason for this situation? Also, the statement is confusing and contradicts an earlier one that sought to state that nitrate concentration was high in the wet season due to increased precipitation. Please check and make the necessary corrections.

All the captions for figures and tables need detailed descriptions.

Other minor comments are:
-Title: I believe the title is too long “Hydrochemical properties, influencing factors, water quality evaluation, and nitrate-related health risk assessment of groundwater in the Baiquan basin, Northern China” For me I think the interest of the paper lies in documenting hydrochemical status of the groundwater and its associate nitrate health risk. It could be rephrased as (this is just a suggestion) “Comprehensive Analysis of Groundwater Hydrochemistry and Nitrate Health Risks in the Baiquan Basin, Northern China”

Experimental design

All are given in the Basic and other comments

Validity of the findings

see additional comments

Additional comments

Here are a few comments to improve the paper:

Line 108 -141: In the study area section, none of the statements were referenced. Could you please clarify whether the information presented is original or based on existing research? If the latter, kindly provide relevant citations to support the claims.

Line 154: institute should start with a capital letter (Geo-mineral Engineering Exploration institute).

Line 294 – 295 The sentence “The excess Ca2+ might be derived from the dissolution of carbonate minerals or Ca-containing silicate minerals”. Are these minerals in the study area? You need to authenticate this sentence by stating the typical carbonate and silicate minerals in the study area that are contributing to these ions in the area. Provide appropriate references of works on the geology of the area to support the claim. Again, in Line 301, is calcite found in the study area? What is the justification for it? Same comments for Lines 309 – 312.

Line 325 – 326 As illustrated in Figure 8a, the distribution of water sample points near the (Na++K+-Cl⁻)/(Ca2++Mg2+-HCO3⁻-SO42⁻)=⁻) = 1. Is it -1 or 1? Please show the equation on the figure.


Line 204-205: Could there be a reason for higher K+, HCO3, NO3-, and F- during dry season than in the wet season?

Line 351: check the sentence: is it rational development or you mean national development?

Line 354 – 356: What guideline values did you use for comparison? Please state the guideline values and support it with appropriate reference.

Line 360 – 367 What is the reason for the disparity in the dry and wet seasons? Again, for Fig. 10, what is the reason for the distribution observed across the study area? It will be good to explain beyond reasonable doubt what could be causing the high values in certain parts and low in other areas of the study are? Ensure you support your claims with relevant literature.

Line 405 – 407: Relate this statement with activities and nature of the geology of the study area and support it with relevant literature of similar situations elsewhere. Line 409 – 411 Give reasons for these occurrences observed. For instance, why is the non-carcinogenic risks predominantly concentrated in the northwest and southeast regions of the study area and not the other parts of the study area? Give reasons for every claim you make and support it with relevant literature.

Line 411 – 413: The sentences here are a bit confusing. Wet season elevated compared to dry season. And later mentioned 65% in dry season and 56% in wet season. It is contradictory.

Line 430 - 432: Conclusion: The 3rd and 4th findings were not well captured in the introduction, especially at the objectives.
What is the future direction of this research? It will be good to state some recommend at the Conclusion Section too.

Lines 226 -228 and 436 – 437: concentrations of K⁺, HCO₃⁻, NO₃⁻, and F⁻ are higher during the dry season compared to the wet season. What could be the reason for this situation? Also, the statement is confusing and contradicts an earlier one that sought to state that nitrate concentration was high in the wet season due to increased precipitation. Please check and make the necessary corrections.

All the captions for figures and tables need detailed descriptions.

Other minor comments are:
-Title: I believe the title is too long “Hydrochemical properties, influencing factors, water quality evaluation, and nitrate-related health risk assessment of groundwater in the Baiquan basin, Northern China” For me I think the interest of the paper lies in documenting hydrochemical status of the groundwater and its associate nitrate health risk. It could be rephrased as (this is just a suggestion) “Comprehensive Analysis of Groundwater Hydrochemistry and Nitrate Health Risks in the Baiquan Basin, Northern China”

---

## Round 0.3 · accepted · Accept

· Academic Editor

Accept

After assessing the revised manuscript, the authors have addressed all the reviewers' comments. The manuscript has been significantly improved and can be accepted for publication.